# Contributions of the Left and the Right Hemispheres on Language-Induced Grip Force Modulation of the Left Hand in Unimanual Tasks

**DOI:** 10.3390/medicina55100674

**Published:** 2019-10-06

**Authors:** Ronaldo Luis da Silva, Francielly Ferreira Santos, Isabella Maria Gonçalves Mendes, Fátima Aparecida Caromano, Johanne Higgins, Victor Frak

**Affiliations:** 1Faculté des Sciences, Université du Québec à Montréal - 141 Avenue du Président-Kennedy, Montréal, QC H2X 1Y4, Canada; frak.victor@uqam.ca; 2Centre de recherche interdisciplinaire en réadaptation (CRIR), Institut universitaire sur la réadaptation en déficience physique de Montréal (IURDPM) - 6300 Avenue de Darlington, Montréal, QC H3S 2J4, Canada; johanne.higgins@umontreal.ca; 3Centro Estadual de Reabilitação e de Readaptação Dr Henrique Santillo – CRER - Av. Ver. José Monteiro, 1655 - Setor Negrão de Lima, Goiânia, GO 74653-230, Brazil; francielly.fisio2010@gmail.com (F.F.S.); isabellaclinicaser@gmail.com (I.M.G.M.); 4Laboratory of Physical Therapy and Behaviour, Department of Physical Therapy, Speech and Occupational Therapy, University of São Paulo Medical School - Rua Cipotânea, 51 - Cidade Universitária, São Paulo, SP 05360-000, Brazil; fcaromano@uol.com.br; 5École de Réadaptation, Faculté de Médecine, Université de Montréal - 7077 Avenue du Parc, Montréal, QC H3N, Canada

**Keywords:** grip force modulation, embodied language, left hand, right hemisphere, left hemisphere, unimanual task

## Abstract

*Background and Objectives:* Language-induced grip force modulation (LGFM) can be used to better understand the link between language and motor functions as an expression of embodied language. However, the contribution of each brain hemisphere to LGFM is still unclear. Using six different action verbs as stimuli, this study evaluated the grip force modulation of the left hand in a unimanual task to characterize the left and right hemispheres’ contributions. *Materials and Methods:* Left-hand LGFM of 20 healthy and consistently right-handed subjects was evaluated using the verbs “to write”, “to hold”, “to pull” (left-lateralized central processing actions), “to draw”, “to tie”, and “to drive” (bihemispheric central processing actions) as linguistic stimuli. The time between the word onset and the first interval of statistical significance regarding the baseline (here as reaction time, RT) was also measured. *Results:* The six verbs produced LGFM. The modulation intensity was similar for the six verbs, but the RT was variable. The verbs “to draw”, “to tie”, and “to drive”, whose central processing of the described action is bihemispheric, showed a longer RT compared to the other verbs. *Conclusions:* The possibility of a given manual action being performed by the left hand in consistent right-handers does not interfere with the occurrence of LGFM when the descriptor verb of this action is used as a linguistic stimulus, even if the possibility is remote. Therefore, LGFM seems to mainly rely on the left hemisphere, while a greater activation of the right hemisphere in action processing appears to slow the increase in LGFM intensity.

## 1. Introduction

In the early 1860s, Paul Broca’s studies highlighted the role of the left inferior frontal gyrus and deemed it the “seat of language” [1], laying a foundation that numerous studies have built upon, emphasizing the left hemisphere’s dominance in linguistic processing [2,3]. In addition to this, dominance of the left hemisphere in motor control has also been extensively studied and documented [4,5,6]. This dual dominance role played by the left hemisphere supports the theory of motor cognition, for which language comprehension and movement production share processing centers and are highly coupled [7]. Furthermore, centers such as Broca’s area, the premotor cortex, and the inferior parietal cortex have been related to both linguistic processing and movement control [8,9].

The assessment of right-hand grip force modulation induced by linguistic stimuli designates a technique which evaluates involuntary grip force modulation due to hearing of manual action verbs. These verbs may be conjugated in sentences or isolated in word lists. Therefore, we have come to understand that the assessment of right-hand language-induced grip force modulation (LGFM) is an excellent tool for the study of motor cognition [10,11,12].

However, da Silva et al. [13] found that left-hand grip force in a unimanual task is likewise modulated by hearing manual action verbs. For these authors, the lower intensity and longer time needed to establish the significance of left-hand LGFM compared to right-hand LGFM seemed to indicate that both hemispheres of the brain contribute to this finding. Assessment of left-hand LGFM is particularly interesting in cases where left hemisphere injury has led to an inability to assess right-hand LGFM.

Manual action verbs usually describe actions unevenly performed by each hand. Oldfield [14] noted “writing” and “drawing” as activities exclusively performed by the right hand of those consistently right-handed. In these cases, verbs such as “to write” and “to draw” do not express an action performed by the left hand. The cortical activation when writing is left-lateralized and this lateralization is more evident in the frontal cortex [15,16]. Cortical activation is more symmetrical when drawing [15], with a more intense activation of right hemisphere regions related to language when compared to writing [16]. Some verbs express actions carried out with either hand, such as “hold” and “pull,” whose central processing is also left-lateralized [17,18], while other verbs describe coordinated asymmetric bimanual activities such as “tie” and “drive”, whose central processing is roughly symmetrical [19,20,21]. Consequently, although all these manual action verbs describe activities performed by right-handers’ right hand, the possibility that they describe an activity performed by the left hand is variable. However, no study has thus far evaluated the effect of these verbs on LGFM of the left-hand.

The LGFM evaluation of these verbs might therefore contribute to elucidating the role of each brain hemisphere in LGFM, as well as in the linguistic processing. The objective of this study was to characterize the contributions of the left and right hemispheres on LGFM by means of six verbs describing actions of variable application to the left hand in a unimanual task.

## 2. Materials and Methods

### 2.1. Ethical Statement

The project was approved by the Université du Québec à Montréal, Canada. Ethical approval was obtained from the Universitary Center of Brasília - UNICEUB Research Ethics Committee (CEP-UNICEUB), Brasília, Brazil—Report no. 2.044.460/17 on 3 April 2017.

### 2.2. Subjects

Twenty healthy subjects participated in the experiment. Subjects were consistent right-handers according to the Edinburgh Handedness Inventory (EHI > 80) [14], with no deficits in cognitive or motor skills, nor neurological or musculoskeletal disorders. They should have at least five years of schooling and be able to read and write. In addition, they should know how to drive a vehicle with a manual gearbox. This last criterion was chosen with regard to the verb “to drive”, which was chosen as a verb describing an asymmetric bimanual function. Informed consent for participation in the experiment was obtained from all participants.

### 2.3. Grip Force Assessment

Participants remained comfortably seated and kept the left forearm supported on a table from the elbow to the distal end of the fifth metacarpal, in a neutral position. Using a three-digital pinch, they were told to hold the grip force sensor and keep it beyond the edge of the table, without support. They were asked to stay as relaxed and as still as possible. A laptop screen was used to show them the variation of the force exerted on the sensor and they were trained to exert a force between 1.5 N and 2.0 N to prevent slippage of the grip force sensor. Participants were asked to maintain a constant level of grip force during the experiment, with no visual feedback of the generated force, since they kept their eyes closed. This protocol has been previously analyzed and described by Nazir et al. [22].

Experimentation was composed of six tests. In each test, the participant listened to a playlist through headphones, lasting about two minutes and divided into two blocks. Each block contained 35 nouns unrelated to a manual action, such as “plane” and “frog”, and a variable number of repetitions of one of the six given action verbs: “to write”, “to draw”, “to hold”, “to pull”, “to tie”, and “to drive” in Brazilian Portuguese. There were a total of 18 repetitions of the action verb by the playlists. These repetitions were interspersed in the word sequence to prevent a sequential presentation or an identifiable distribution scheme. The interval between two consecutive words was 1000 ms. The six playlists can be found at the Appendix A: Playlists. The list of words and their equivalents in English is provided in the Appendix A: Word list. The action verb was presented to the participant as a “keyword”, without drawing attention to its grammatical class. Before the test, the participant was instructed to mentally count the number of keyword repetitions, in order to keep their attention on the current language stimulation. By finishing the first block of listening, they opened their eyes, put the grip force sensor on the table and reported the number of repetitions. The procedure was repeated for the second block following a one-minute rest. They had two minutes of rest by the end of the test, and a new test was performed with a second keyword. The experiment’s sequence of keywords was randomly defined. The study dataset can be found in the Appendix A.

The force sensor was connected to an amplifier (Honeywell DV10L) that was connected to an acquisition card (Measurement Computing USB-1608GX). The compression force was recorded in mN/ms with 1 kHz data transmission for a laptop. Dasylab 11.0 software was used to filter the data at 15Hz, by means of a fourth-order Butterworth zero-phase low-pass 50 Hz band-drop filter to display the force variation. The laptop also sent the playlists to the acquisition card, which were then delivered to the participants by the headphones.

### 2.4. Data Analysis

The data were segmented from 200 ms before the keyword onset till 1000 ms following its beginning. The 200 ms average signal before the start of the action verb (baseline) was used to normalize the data for that verb, and this procedure was repeated for each occurrence. If the signal variation between 200 ms before and 800 ms following the word onset was greater than or equal to 200 mN, the data for that occurrence were rejected. In the same way, the data of an occurrence were rejected if there was a force increasing at a rate greater than 100 mN within 100 ms [22]. If more than 30% of the repetitions of a verb were rejected, data from that participant were rejected. Since the comparison between non-action nouns and action verbs is currently well documented [10,11,12], only the action verbs were analyzed in this study.

The baseline was compared to the three phases of linguistic processing defined by Friederici [23] by means of one-factor repeated measure ANOVA to evaluate the occurrence of LGFM for each verb. According to Friederici, the analysis of the syntactic structure characterizes Phase 1 (100–300 ms). Phase 2 (300–500 ms) presents a broader analysis that includes lexical-semantic and morphosyntactic processes, and during Phase 3 (500–800 ms) the information generated in the previous phases is reanalyzed and integrated. The occurrence of LGFM was defined as a significant increase in grip force between the baseline and one or more of these phases [13]. A one-factor repeated measure ANOVA was used to evaluate the occurrence of LGFM using the mean LGFM value of each participant. Tukey’s post hoc test (DSH) was used to identify the phase in which the grip force became significantly different from the baseline. Data from this phase was reordered in 50 ms time intervals and a new one-factor repeated measure ANOVA was performed to identify the first significant time interval. Data of the selected time interval and its predecessor were reorganized in 10 ms micro-intervals and a new one-factor repeated measure ANOVA was performed to identify the first significant micro-interval. The time between the word onset and the first micro-interval significantly different from baseline was called the reaction time (RT). The RT was identified for each of the six verbs.

Secondly, the action verbs were paired according to their possibility of functional application on the left hand. The verbs “to write” and “to draw” formed the pair of non-applicable action verbs, “to hold” and “to pull” made up the pair of optional action verbs, and “to tie” and “to drive” the pair of shared action verbs. To compare the three action verb pairs, three two-factor ANOVA with repeated measures were conducted, one for each linguistic processing phase.

Lastly, the action verbs “to write”, “to hold”, and “to pull” were compared as a group to the verbs “to draw”, to tie”, and “to drive” as a second group. The first group was composed of verbs expressing actions whose central processing is left-lateralized (LHCP), while the central processing in the second group involves a significant participation of the right hemisphere (RHCP). These groups were also compared by means two-factor ANOVA with repeated measures, phase by phase. A Spearman correlation test was used to evaluate the correlation between sex, age, years of schooling, manuality score, and LGFM for each of the six verbs. In this case, the median values of the time intervals for each subject were used to make the comparisons. Spearman’s correlation test was chosen based on sample size and non-normal distribution of data, when it was indicated by the Shapiro-Wilk test.

## 3. Results

### 3.1. Subjects

Twenty healthy subjects aged 20 to 55 years (7 women, 31.1 ± 8.8 years old and 13 men, 31.1 ± 9.7 years old) participated in this study, but two data subjects were excluded from the analysis for having lost five of the six verbs. They had between 5 and 18 years of schooling (women from 8 to 18 years, mean 12.3 ± 3.7 years, men aged 5 to 15 years, mean 11.7 ± 2.7 years) and scored 80 to 100 according to the Edinburgh Handedness Inventory (EHI) as consistent right-handers (women 85 to 100, mean 88.5 ± 7.2, men 80 to 100, mean 94.3 ± 5.3).

### 3.2. Language-Induced Grip Force Modulation Occurrence

The six verbs produced LGFM. RTs were found in Phase 1 of linguistic processing for the verbs “to write” and “to pull” and in Phase 2 for the verbs “to hold”, “to draw”, “to drive”, and “to tie”. The verb “to write” presented the lowest RT (250–260 ms) and the verb “to tie” presented the highest (410–420 ms). Table 1 presents the findings related to LGFM occurrence, as well as the RT determination.

### 3.3. Comparisons by Verb, Verbal Category, and Hemispheric Processing

According to a two-way ANOVA with repeated measures performed with the six action verbs, the LGFM was similar among all of them (F _(5,102)_ = 0.438; *p* = 0.8209). Figure 1 presents the curves of each action verb along the linguistic processing phases. The analysis performed with the functional application groups did not find statistical differences between the three pairs of action verbs, as shown in the Figure 2.

LGFM was significantly more intense (F _(1,106)_ = 5.700; *p* = 0.0187) for LHCP than RHCP during Phase 1 of the linguistic processing. There was no statistical difference along the other phases (Phase 2: F _(1,106)_ = 0.002; *p* = 0.9641); Phase 3: F _(1,106)_ = 0.144; *p* = 0.7049) (Figure 2).

### 3.4. Correlation Analysis

The correlation analysis between the LGFM and sex, age, and years of schooling did not find significant relationship (*p* > 0.05). The values of *r* and *p* for these correlations are presented in Table 2.

## 4. Discussion

In this study, six verbs were chosen as linguistic stimuli to evaluate the language-induced grip force modulation of the left hand in a unimanual task. These verbs were grouped into three categories related to the use of the left hand by consistent right-handers: non-applicable action verbs, optional action verbs, and shared action verbs.

The six action verbs were able to modulate grip strength. Thus, the fact that it was a non-applicable, optional, or shared action verb did not prevent any of the verbs from modulating the grip force. Additionally, there was no difference among the modulation curves, so that the nature of the action described by the verb did not influence the intensity of the modulation. The modulation of the grip force, therefore, seems to be independent of the possibility that the action is performed by the left hand, or that the left hand is performing either alone or is asymmetrically accompanied by the right hand.

The understanding of the action seems to be related to the motor centers’ activation, irrespective of the immediate execution potential or the choice regarding the member that will act. Thus, language-induced grip force evaluation is an effective way to evaluate the connection between language processing and motor control, irrespective the hand chosen for evaluation.

Although manuality and language are lateralized in early childhood [24,25,26], the non-dominant hand is not detached from the connection between language and motor processing. Our finding therefore suggests that this connection should have early development and little influence from one’s environment [24,25].

The analysis of the reaction time of the six verbs, however, found differences between them. The verbs “to write” and “to pull” presented the lowest RT, being the only ones to have their RT in the first phase of the linguistic processing. The shared action verbs “to tie” and “to draw” displayed the highest reaction times. As they describe a bimanual action, although asymmetric, it is expected that the right hemisphere is more activated for these verbs when compared to the right hemisphere activation for writing [19,20,27,28]. Furthermore, drawing showed the same RT as “to drive”, and the literature describes greater activation of the right hemisphere for carrying out this task. Thus, the RT analysis seems to indicate that the greater involvement of the right hemisphere in accomplishing the task described by these action verbs negatively contributes to growing modulation throughout the first phase of linguistic processing. However, the verb “to write”, whose activation is described as essentially left-lateralized [29], exhibited the lowest reaction time. In our study, two verbs demonstrated lower RT than that observed for left hand unimanual condition by da Silva et al. [13], also, one verb had a similar RT and three others presented higher RTs. The six verbs’ average RT of 326.7 ms falls exactly in the range described in that article, that is 320–330 ms. The RT of the LGFM of the left hand in unimanual activity is therefore a direct function of the chosen action verb, according to the participation of the right hemisphere on its linguistic processing.

The optional action verbs showed modulation curves very similar to the curve of the verb “to write” throughout the first phase of linguistic processing (100–300 ms). In fact, the ANOVA performed with the verbs divided into two blocks, left-lateralized central processing action verbs versus non-lateralized central processing action verbs, detected a significant difference between them. This difference disappears in the second phase of language processing, when all verbs reach their RT. Although there is no study investigating the zones activated by “to hold” and “to pull”, these verbs are optional verbs, like “to grip” and “to grasp”, and these actions are described as components of gripping and grasping actions [18,30]. Therefore, holding and pulling should rely on the gripping circuit, which is widely described as left-lateralized [4,8,31].

As the modulation intensity was not affected by the type of hemispheric treatment, our findings seem to indicate that the intensity of LGFM is mainly determined by the left hemisphere. The influence of the left hemisphere on the right hemisphere, necessary for the modulation to be expressed by the left hand, may be due to the action of the Broca’s area on its counterpart in the right hemisphere [32], or the action of the Broca’s area on the homolateral ventral premotor cortex [33,34], which maintains a strong connection with its contralateral counterpart [34]. However, the effect of greater activation of the right hemisphere on the reaction time of bihemispherically processed verbs seems to indicate that the Broca’s area influences the right hemisphere primarily by the ventral premotor cortex.

The participation of the Broca’s area homolog in normal language processing is not yet clear [35,36]. Some studies have shown that activation of the Broca’s area homolog may be related to syntactic reanalysis and sentence restructuring, rather than to normal syntactic linguistic treatment [37,38,39]. In this case, the action of the Broca’s area homolog seems to require a longer processing time [40]. Our study found that the right hemisphere slowed the grip force increase, so that the reaction time of the bihemispheric processing verbs was longer than those of the left-lateralized processing verbs. Our findings seem to indicate, however, that the Broca’s area homolog may be involved in normal linguistic processing. In this case, unilateral left-hand motor activity may be an extra stimulus for Broca’s area homolog engagement [29]. This hypothesis deserves further attention, as it supports the notion that unilateral simultaneous motor recruitment can be an important strategy for approaches involving linguistic processing and production.

## 5. Conclusions

In short, the relevance of the action described by the verb to the member under evaluation does not seem to influence either the production of the LGFM or the intensity of the modulation. A greater involvement of the right hemisphere in the central processing of the action seems to imply in a longer time for the modulation to become significant in comparison to the actions with left-lateralized central processing. Therefore, it can be concluded that both the left hemisphere and the right hemisphere contribute to the production of the language-induced grip force modulation of the left hand in a unimanual task.

## Figures and Tables

**Figure 1 medicina-55-00674-f001:**
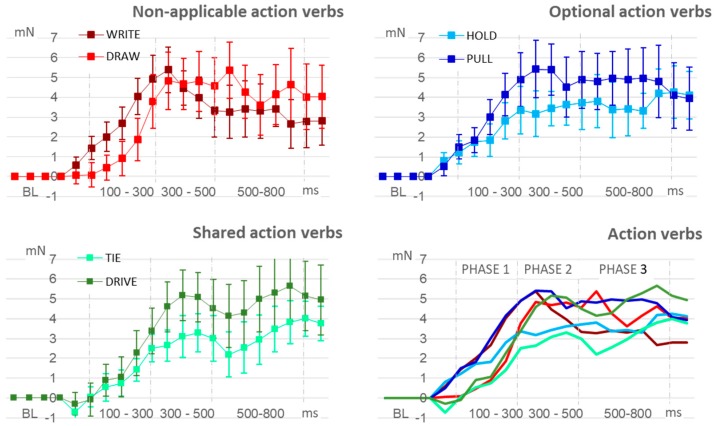
Language-induced grip force modulation by action verb group. Phases 1, 2, and 3 represents the linguistic processing phases described by Friederici [23]. mN: grip force modulation in millinewtons. ms: time-interval in milliseconds.

**Figure 2 medicina-55-00674-f002:**
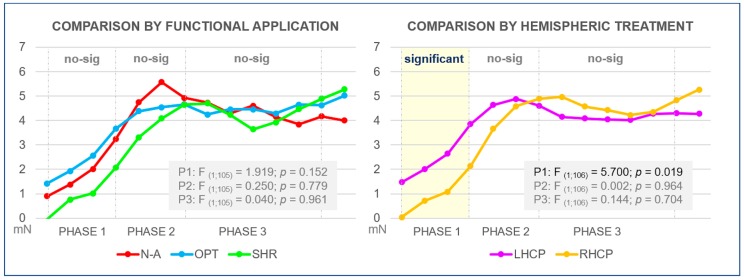
Comparison between. Phases 1, 2, and 3 (P1, P2, and P3, respectively) represents the linguistic processing phases described by Friederici [23]. N-A: non-applicable action verbs. OPT: optional action verbs. SHR: shared action verbs. mN: grip force modulation in millinewtons.. LHCP: action verbs with left-lateralized central processing. RHCP: action verbs whose central processing comprises an important participation of the right hemisphere. No-sig: no statistically significant difference between LHCP and RHCP or among N-A, OPT, and SHR curves.

**Table 1 medicina-55-00674-t001:** Statistical notation of language-induced grip force modulation occurrence and reaction time determination by interval and micro-interval.

Verb		Result	Notation
Write	LGFM	+	F _(1.77,30.089)_ = 8.345, *p *= 0.002
	interval (ms)	250–300	*q*_(289)_ = 5.782, *p *= 0.007, *d *= 1.36
	RT (ms)	250–260	*q*_(153)_ = 5.194, *p *= 0.0119, *d *= 1.22
Draw	LGFM	+	F _(1.824,31.006)_ = 9.101, *p *= 0.001
	interval (ms)	350–400	*q*_(289)_ = 5.954, *p *= 0.0044, *d *= 1.4
	RT (ms)	350–360	*q*_(153)_ = 4.991, *p *= 0.0191, *d *= 1.18
Hold	LGFM	+	F _(1.446,24.578)_ = 6.464, *p *= 0.010
	interval (ms)	300–350	*q*_(289)_ = 5.413, *p *= 0.0181, *d *= 1.28
	RT (ms)	330–340	*q*_(153)_ = 4.668, *p *= 0.0385, *d *= 1.1
Pull	LGFM	+	F _(2.022,34.373)_ = 15.619, *p* < 0.001
	interval (ms)	250–300	*q*_(289)_ = 6.164, *p *= 0.0024, *d *= 1.45
	RT (ms)	270–280	*q*_(153)_ = 4.978, *p *= 0.0196, *d *= 1.17
Tie	LGFM	+	F _(1.679,28.549)_ = 9.222, *p *= 0.001
	interval (ms)	400–450	*q*_(289) _= 5.283, *p *= 0.0249, *d *= 1.25
	RT (ms)	410–420	*q*_(153)_ = 4.883, *p *= 0.0243, *d *= 1.15
Drive	LGFM	+	F _(1.586,26.956)_ = 11.155, *p *< 0.001
	interval (ms)	350–400	*q*_(289)_ = 6.586, *p *= 0.0007, *d *= 1.55
	RT (ms)	350–360	*q*_(153)_ = 5.415, *p *= 0.007, *d *= 1.28

LGFM: language-induced grip force modulation; RT: reaction time; ms: milliseconds.

**Table 2 medicina-55-00674-t002:** Spearman’s correlation, grip force modulation and sample characteristics.

	Write		Pull		Hold		Draw		Tie		Drive	
	*r*	*p*	*r*	*p*	*r*	*p*	*r*	*p*	*r*	*P*	*r*	*p*
Sex	0.26	*0.312*	−0.47	*0.115*	−0.29	*0.115*	−0.29	*0.115*	0.16	*0.762*	−0.31	*0.232*
Age	−0.02	*0.958*	0.33	*0.197*	−0.04	*0.598*	−0.04	*0.598*	−0.17	*0.810*	0.41	*0.233*
Schooling	0.03	*0.716*	0.44	*0.280*	0.22	*0.362*	0.22	*0.362*	0.04	*0.673*	0.05	*0.570*
Consistence ^1^	−0.19	*0.443*	0.40	*0.169*	0.12	*0.492*	0.12	*0.492*	−0.40	*0.113*	0.38	*0.112*

^1^ Consistence refers to handedness consistence according to Oldfield [11].

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
