# Peer review of "Contributions of the Left and the Right Hemispheres on Language-Induced Grip Force Modulation of the Left Hand in Unimanual Tasks"

_medicina, 2019, doi:10.3390/medicina55100674_

Round 1
Reviewer 1 Report
The object of the study is interesting and according to the authors, it is a pioneer in the evaluation of the effects of selected verbs in LGFM. The results presented seem to open opportunities for numerous future studies for a more detailed understanding of the neurofunctional circuitry involved in LGFM.
My overall impression shows that the style of the text requires revision to become more scientific and less colloquial. However, the information contained in the text is adequate, although it should be more detailed in relation to literature data. It seems to me that the authors were very economical and brief in writing the introduction and discussion and in confronting their findings with the literature.
Still, I suggest reviewing the writing style.

Author Response
We appreciate your consideration of our article. We highlight your suggestions for answering each one below.
My overall impression shows that the style of the text requires revision…Following proceeding with the adjustments requested by the review, we submitted the article to a professional proofreading service. We therefore believe that our article now has a better level of writing.
… the information contained in the text … should be more detailed in relation to literature dataThank you for this consideration. We improved the introduction by including our study in the theory of motor cognition and pointing its contribution to this theme. Six references were exchanged, and three references present in the discussion were shifted to the introduction to better fit the new design. These changes also enabled a better integration between both sessions. In addition, the discussion was deepened in the light of the literature. The added excerpts are highlighted in the file Manuscript- modifications.
… authors were very economical and brief in writing the introduction and discussion…We apologize for that. Medicina is a journal that does not impose word limits on submitted articles, relying on the authors’ knowledge and common sense to write a text that is both comprehensive and concise. In this way, we were able to improve the introduction and the discussion to better demonstrate the scientific value of our study.
… and in confronting their findings with the literatureWe made several modifications throughout the article so that our discussion could clearly show where our study falls within the theory of motor cognition and the contributions it can make to discussions on this topic. Thus, the discussion session was deepened in its reasoning and in its connection with the current knowledge brought by the literature. We therefore believe that our discussion is now more consistent and

Reviewer 2 Report
In the method section sentences like: "they open their eyes..." occur. "They" should be replaced by more formal noun, e.g. participants In data analysis and statistical analysis sections the information are similar - authors should unit both sections. On Fig. 1 It is not clear what colorful dots represents - it should be described or present in other way Fig.1 - data presented on figure are clearly data points and should be presented as points, not connected lines (or point connected with lines). It is also not clear if presented data is mean or median (authors says about both). If it's mean, standard deviation should be included. Authors say that they registrated grip force modulation, when participants listen to noun words and action word, but the results of noun word were not analysed. I think that analysis of this part is essential and is important to show contrast between both kinds of words. Authors should show baseline during 200 s (not as a point) to show that in control condition there is no such variation as after words listening The comparison between "to write", "to pull" and "to tie" and "to draw" should be presented with statistical analysis in form of other figure or table. Manuscript should be checked in the context of minor mistakes (e.g. small letter starting first paragraph of Conclusions)Author Response
We appreciate your consideration of our article. We highlight your suggestions for answering each one below.
… more formal noun …Following proceeding with the adjustments requested by the reviews, we submitted the article to a professional proofreading service. “Language-induced GFM” was changed to “LGFM” following your suggestion. We therefore believe that our article now has a better level of writing.
… data analysis and statistical analysis sections … - authors should unit both sectionsWe merged these sessions.
1 It is not clear what colorful dots represents - it should be described or present in other wayColorful dots were used to indicate the RT for each verb. We adopted a different system to indicate the RT in the graph.
1 - data … should be presented as points, not connected lines (or point connected with linesWe changed the figure 1 to show points connected with lines with error bars.
It is also not clear if presented data is mean or medianFigure 1 and all ANOVAs was based on the mean values presented by the group. We included a phrase in the text to clarify it. Spearman correlations (Table 2) was performed using mean and median individual values. Since the results was quite similar by both methods, we opted to present the Spearman correlations based on the median LGFM individual values. We included in the text this consideration aiming to clarify this choice.
If it's mean, standard deviation should be includedError bars were included in a new Figure 1.
… the results of noun word were not analysed. I think that analysis of this part is essential and is important to show contrast between both kinds of wordsAccording to articles previously published in this field of ​​study, verbs present a significantly more intense modulation than nouns. This comparison was made in studies that used words or phrases as linguistic stimuli. Thus, we believe that this relationship is well established in the literature. The aim of our study was to evaluate whether verbs describing actions with variable potential achievement by the left hand could modulate the grip force of this hand in unimanual activity. In left unimanual activity, the influence of the left hemisphere on the activity of the left hand is significantly reduced, while the right hemisphere participation becomes more significant.
Nouns were used as a background in the playlists so that the study could have a greater methodological similarity to other studies in the literature and thus could be compared to them. For nouns to be evaluated, it would be necessary to define some of them as a keyword, using verbs for the background. This would make the evaluation very extensive and exhausting for participants and could endanger the quality of the study.
The evaluation of the noun-induced left-hand modulation, however, may yield interesting information, which motivates us to integrate such questions into our laboratory's future research plans.
Authors should show baseline during 200 s (not as a point) to show that in control condition there is no such variation as after words listeningWe appreciate your suggestion. Baseline was obtained by an average of the 200 ms normalized data. The values are extremely close to zero such that the values given in the supplementary material are described in scientific notation. Standard error values have the same exponential basis, which is why there is no error bar for points before the beginning of the word. However, since each point on the graph represents the time interval of 50 ms, we represent the period before the word starting with four points, which correspond to the period used to determine the baseline.
The comparison between "to write", "to pull" and "to tie" and "to draw" should be presented with statistical analysis in form of other figure or tableWe appreciate your suggestion. A new figure was designed to illustrate the comparisons between the pairs of action verbs (nonapplicable, optional or shared action verbs) and between the groups of verbs by hemispheric treatment.
Manuscript should be checked in the context of minor mistakes (e.g. small letter starting first paragraph of Conclusions)We than you for this consideration. The text has been rechecked following the professional proofreading for minor mismatch and formatting errors.
